# Prioritizing the Risk Factors of Severe Early Childhood Caries

**DOI:** 10.3390/dj5010004

**Published:** 2017-01-06

**Authors:** Noha Samir Kabil, Sherif Eltawil

**Affiliations:** 1Department of Pediatric Dentistry and Dental Public Health, Faculty of Dentistry, Ain Shams University, Cairo 11566, Egypt; 2Department of Pediatric Dentistry and Dental Public Health, Faculty of Oral & Dental Medicine, Cairo University, Giza 12613, Egypt; Sherif.eltaweil@dentistry.cu.edu.eg

**Keywords:** severe early childhood caries, prioritizing risk factors

## Abstract

Severe early childhood caries remains the most common chronic disease affecting children. The multifactorial etiology of caries has established a controversy about which risk factors were more significant to its development. Therefore, our study aimed through meticulous statistical analysis to arrange the “well agreed upon” common risk factors in order of significance, to aid the clinician in tailoring an adequate preventive program. The study prioritized or reshuffled the risk factors contributing to severe early childhood caries and placed them in the order of their significance as follows: snacking of sugary food several times a day, increased number of siblings to three or more, night feeding, child self-employed brushing, mother’s caries experience, two siblings, on demand feeding, once/day sugary food, sharing utensils, one sibling, male gender, father’s education, late first dental visit, brushing time, mother’s education, no dental visit, decreased brushing frequency, and no night brushing.

## 1. Introduction

Early childhood caries (ECC) remains a major unresolved dental public health problem in developing as well as developed countries, despite the continuous trials for implementation of preventive strategies [1,2]. The decline in the prevalence of ECC among children in developed countries cannot be denied but it continues to progress at epidemic proportions in low-middle income countries [3,4].

It has been established in the literature, that any child younger than six years of age presenting with at least one tooth which is decayed (cavitated or non-cavitated), missing (due to caries), or filled tooth surfaces in any primary tooth is suffering from ECC. When the child is younger the condition is more severe and hence the nomenclature “Severe early childhood caries (S-ECC)” [5].

Besides having a higher risk of developing new carious lesions at adolescence and adulthood, children with untreated and neglected S-ECC are prone to complications such as pain, absence from school days, compromised eating habits, and low self-esteem. These consequences will all adversely affect the children’s well-being and their oral health-related quality of life (OHQoL) [6,7]. Moreover, the estimated treatment costs and expertise of highly skilled professionals required for treatment of the disease places pressure on the economy, which made it necessary to trace the contributing risk factors of early childhood caries in order to tailor a preventive economic approach to combat the deep-rooted disease [7].

Unfortunately, the risk indicators for the presence of dental caries in young children are far from a handful; in a systematic review, almost 90 risk factors were described [8]. This complex etiology of S-ECC has intrigued researchers to dig deeper into the various risk factors involved besides feeding [9] and oral hygiene practices [10]. Other contributing factors included, but were not limited to; *Streptococcus mutans* levels, active dental problems in parents/caregivers, socioeconomic status, and the onset of the first dental visit [11,12,13], which all apparently contributed to the risk of developing the disease [8].

Within the limitations of variations and confounders from different samples, as well as the study design itself, the statistical analysis techniques may also lead to different conclusions. As it would be impossible to ignore the multifactorial nature of ECC, and as far as our knowledge there was no study in the literature that places the risk factors in order of significance, our study therefore aimed through meticulous statistical analysis to arrange the “well agreed upon” common risk factors in order of significance in order to aid the clinician in tailoring an adequate preventive program for young children. We present it as an attempt to prioritize or reshuffle the risk factors contributing to severe early childhood caries and place them in the order of their significance.

## 2. Materials and Methods

### 2.1. Sampling and Sample Size

A minimal sample size of 108 was calculated using EpiCalc 2000 version 1.02 program (Brixton Health, USA) assuming a power of 80% and alpha = 0.05. It was based on the percentage of oral hygiene practices performed for children (‘no brushing’ and ‘brushing three times/day’ were 31.05% and 12.11%, respectively) [14].

We designed a cross-sectional study to analyze and prioritize the risk factors for ECC among preschool children. Prior to the main study, a pilot study was carried out in the Department of Pediatric Dentistry and Dental Public Health, Faculty of Dentistry, Ain Shams University, Cairo, on a group of 25 children. The preliminary study was carried out to evaluate the feasibility of conduct of a larger study and to aid in the calculation of sample size.

### 2.2. Study Design and Ethical Approval

In considering subject selection, in order to identify children for the study, Cairo was divided into four strata by administrative boundaries (four largest counties in Cairo). Subsequently, three immunization centers were selected in each stratum as primary sampling units and 13 subjects were selected randomly from each primary sampling unit. The project was submitted to and approved by the ethical committee of Ain Shams University. The study was conducted from November 2015 to April 2016. On the days assigned for examination patients attending the facility fulfilling the inclusion criteria were identified, then 13 were chosen with the aid of random tables for each day until the required sample size was reached.

### 2.3. Inclusion and Exclusion Criteria

Inclusion criteria were normal healthy children aged two to four years. Children with serious medical problems, those attending with a caregiver other than their mother whom the authors thought would not give accurate information in the process of interviewing, or children whose mothers declined to participate were excluded. Another exclusion criterion was previous fluoride varnish application as it could affect the bacterial *Streptococcus mutans* levels.

After their routine oral examination, the mothers were approached, the purpose of the interview was clearly explained; and they were assured that there were no possible side effects for the study on their children; after their verbal approval for participation, they were requested to sign the “Informed Consent” prior to enrollment. The final sample comprised 140 mothers and their preschoolers from the 12 assigned immunization centers.

All parents were motivated by offering preventive procedures such as fluoride application, and educational sessions on the proper oral hygiene measures and the needed dental treatment.

### 2.4. Data Collection

The study was divided into three stages: stage 1 included the intra-oral examination, stage 2 was the saliva sample collection, and stage 3 was a questionnaire survey.

#### 2.4.1. Intra-Oral Examination

The children’s teeth were examined by the principal investigator (N.S.K) who was unaware of the outcome of the mothers’ interview by the co-author (S.E).

The intraoral examination was conducted in accordance with WHO (World Health Organization) standards in a well-lit natural light area with the use of disposable plain dental mirrors, gauze wipes, and wooden tongue depressors. Children were examined in a supine position on the examination beds present in the facility, except for those infants who were either required to be held on the lap of their caregiver, or required the “knee to knee” position for examination [3]. A child was considered to be suffering from ECC when he or she complied to the definition of the American Academy of Pediatric Dentistry (AAPD), which defined ECC as the presence of one or more decayed, missing, or filled tooth surface in any primary tooth in a child of 71 months of age or younger, while the term S-ECC is used to describe any sign of smooth surface caries in children younger than three years of age, also from ages three to five, one or more cavitated, missing due to caries, or filled smooth surfaces in primary maxillary anterior teeth or a decayed, missing, or filled score of ≥4 (age 3), ≥5 (age 4), or ≥6 (age 5) surfaces constitutes S-ECC. A white spot lesion or a filled tooth with recurrent caries was considered relevant [5].

#### 2.4.2. Saliva Sample Collection and Estimation of *Streptococcus mutans* (S. *mutans*) Level

Saliva samples were obtained randomly from 25 children suffering from S-ECC, and 15 who were caries-free, as a representative sample for the rest of the population.

Whole saliva was collected between 9.30–11.30 a.m. with the aid of small disposable plastic syringes. The subjects were asked to refrain from eating for one hour before collection. Approximately 2 mL of saliva was collected in a sterile syringe and transported to the microbiology laboratory, then 1 mL aliquot of saliva was transferred to a labeled sterile tube containing 4 mL of broth (thioglycolate broth). The saliva sample was vortexed, to uniformly mix the saliva and the broth, using a cyclomixer.

Using an inoculation loop (4 mm inner diameter), 10 μL of the vortexed 1:5 dilution sample was streaked in duplicate on Mitis salivarius bacitracin agar (MSB) selective for *S. mutans*. The MSB agar plates were incubated anaerobically for 24 to 48 h at 37 °C in 5% CO_2_ in nitrogen. After incubation, counts were done for the colonies with morphological characteristic for *S. mutans* on the MSB agar. Identification for *S. mutans* was confirmed by biochemical tests including manitol fermentation and gram staining catalyst test. Colony counting was done with a magnifying glass, and the count of *S. mutans* was expressed as the number of colony forming units per milliliter (CFU/mL) of saliva. The actual colony count was then multiplied by 1 × 103 on account of the saliva sample having been diluted one thousand times (1:5 dilution) [15,16,17,18].

#### 2.4.3. Oral Health Questionnaire

The mothers were interviewed by the researcher (S.E) who was unaware of the outcome of the children’s oral examination performed by the other researcher (N.S.K).

A structured questionnaire interview was designed after reviewing the literature, to choose from the most common, relevant, and significant risk factors of ECC [8,13,14,19,20]. It was pre-tested for validity, reliability, and clarity in the department, with the aid of 20 volunteer mothers of young children. During the trial, some questions were found to be confusing. These questions were revised and retested with other volunteering mothers. The responses were recorded through direct interviewing.

The questionnaire included four domains; sociodemographic and socioeconomic factors, oral hygiene measures, feeding practices, and dietary habits.

**i Sociodemographici**

Questionnaires were administered to the mothers of participating children to obtain sociodemographic information such as name, gender, age, the number of siblings, level of education of the mothers and fathers, which were categorized into two levels: (a) school only, which included those who had their education limited to school or were school drop outs, and (b) University education or higher degrees. Mothers were asked about the presence of carious cavities or fillings in their mouth; if unsure they were examined for verification. To throw light on the child’s oral health, the mother was asked about any previous dental visit, whether it was early, late, or whether there had not been a previous visit at all.

**ii Oral hygiene practice**

In this domain, mothers were asked about their child’s brushing frequency. Answers were categorized into ‘no brushing’, ‘brushing once or twice/week’, ‘once/day’, and ‘twice/day’. They were specifically asked if brushing was performed at night or any random time. Finally, they were asked if brushing was self-implemented or performed by the mother.

**iii Feeding practice**

Dietary and nutritional information involved questions about whether the child was breast or bottle fed. History of whether nighttime and on demand feedings was practiced. Weaning age of the child and whether the household shared the same utensils, specifically the spoon, was also investigated.

**iv Dietary Habits**

Information was collected about consumption of food containing sugar, which was referred to as “sugary food”, sweets, and drinks as well as fruit juices (answers were categorized into ‘never’, ‘once a week’, ‘two or three times/week’, ‘once/day’, ’two or three times/day’, and ‘several times/day’).

### 2.5. Statistical Analysis

Both the questionnaire and oral examination forms were manually checked for completion of the required information. Data was collected, recorded in a standardized form, and then entered into the SPSS software for statistical analysis.Chi-square test was performed for each categorical variable to assess whether significant differences were observed between the two groups (Figure 1, Figure 2, Figure 3 and Figure 4).Spearman correlation was used to find the correlation between the different risk factors and the log streptococcal count (Table 1).A logistic regression model was used to identify risk factors for caries development among children by considering independent variables simultaneously (Table 2). A *p*-value < 0.05 was considered statistically significant.The odds ratio was calculated for all possible risk factors associated with the prevalence of EEC and arranged in descending order after omitting all the insignificant variables, which was the main idea of the study (Table 3).

## 3. Results

Chi-square was performed on variables in each of the four assigned domains (Figure 1, Figure 2, Figure 3 and Figure 4).

There were statistically significant differences among children with ECC and caries-free children in all socio-demographic data.

Regarding the oral hygiene practices, there was a highly significant statistical difference, between children with ECC and caries-free children in all the studied aspects. In ECC, the higher percentage of children either never brushed or employed self-brushing only once or twice a week.

Regarding the consumption of sugary food and sweets, a highly significant statistical difference existed between the two groups, while none was encountered regarding the consumption of fruit juices.

As displayed in Table 1, there were direct correlations between male gender, mother’s current caries experience, the number of siblings, feeding on demand, night feeding, sharing utensils, sugary food intake frequency, and log. Streptococcal count in both groups.

While there was an inverse correlation between mother’s and father’s education, earlier child first visit, increased brushing frequency, night brushing, who brushes (mother), and log. Streptococcal count.

Possible risk factors of socio-demographic characteristics, feeding practices, and oral hygiene practices were adjusted into a regression model. Number of siblings, who brushes, and sugar consumption variables could not be entered into the regression since one cell of the table was empty.

After adjustment for the other possible risk factors, children who were night feeding, whose mothers had current caries experience, and with male gender predominance, will have a greater chance to develop ECC by 44.48, 29.30, and 11.54 times, respectively, than those who were not night feeding, had caries-free mothers, and a female gender predominance. Children who benefited from early dental visits had less chance to develop ECC than those who never visited the dentist by 93.2%.

To override the omitted variables, finally, the odds ratio was performed for all four domains then combined and arranged in one table (Table 3) in descending order (starting with the highest odds ratio) after omitting the non-significant variables. This table represents the core of the study or it is the “master table” as it displays the various risk factors in order of significance which gave us the opportunity to prioritize these factors instead of just counting them collectively.

Caries was found to develop more in children with one, two, and three siblings than those with zero siblings by a rate of 3.32, 12.67, and 48 times, respectively.

A higher risk of caries was found in children whose mothers had previous caries experience by 20.44 times more than those whose mothers were caries-free.

Children with male gender predominance had a greater chance of 2.91 times more than females to develop caries.

Children who had a late or early first dental visit had less chance to develop caries than those who never had dental visits by 75.2% and 95.8%, respectively. The children whose mothers and fathers had a university education had less chance to develop caries than those whose mother’s and father’s education was limited to school by 91.7% and 61.9%, respectively. Children with bushing frequency twice/day and once/day had less chance to develop ECC than those who never brushed by 96.7%.

Children who performed night brushing or no specified brushing time had less chance to develop ECC than those who never brushed by 97.2% and 89.1%, respectively. The children who were responsible for their own oral hygiene practices had a greater chance to develop ECC by 22.90 times more than those whose mothers were responsible for brushing.

Night feeding, feeding on demand, and sharing utensils was related to a higher chance of developing ECC by 40.05, 9.66, and 8.79 times, respectively, more than those not practicing these habits. Children with the frequency of sugary food intake once/day, two to three times/day, and several times/day had greater chance to develop ECC by 9.0, 76.50, and 135.0 times, respectively, more than those whose frequency of sugary food intake was never.

## 4. Discussion

Severe ECC remains a cause of both social and economic burden throughout the world. It has also been proven to affect the child’s quality of life; therefore, it is mandatory for those interested in the improvement of public health to be cognizant of the deterioration in the level of oral health and increase in dental caries, and to direct their efforts towards the prevention of these problems [21,22].

A deeper insight into the risk factors of ECC is always of value to aid in its prevention and management. As previously mentioned, a systematic review stated that almost 90 risk factors were described for ECC; as such, the authors chose those that were agreed upon and repeated in most of the studies present in the review [8]. The findings of our study agreed with several others in most of the different variables in the tested domains, but our main aim was to prioritize the risk factors in an order of significance in order to shed light on the weight of each isolated variable as a caries risk indicator.

The collection of data on children’s lifestyle factors and sociodemographic factors was performed by direct interviewing of parents rather than having them fill in a questionnaire by themselves in order to avoid misinterpretation of questions, which was noticed during the pilot study. Face to face questionnaires were assumed by some researchers to be “image enhancing” because subjects will attempt to say what they knew was right, rather than what they actually practiced [23], yet the low educational and socioeconomic level of many of the patients examined, made it impossible to conduct a self-administered questionnaire.

An age group of two to four years was chosen due to the difficulty of finding a caries-free cohort older than this age, as revealed from the results. The mean age of children with S-ECC was found to be higher than the caries free group. This finding was consistent with several studies that related this to the fact that older children have been exposed to cariogenic challenges for a longer time [8,10]. Further explanation to this can be attributed to the fact that as the age of the child increases the nature of food consumption differs (i.e., as children grow older especially at school age, they tend to consume more sweets).

Based on the clinical examination of children, the mean value of DMFT (dental caries index) in children with S-ECC was found to be 9.96 ± 3.86, which was close to other studies in developing countries [24,25].

In our study, boys have been reported to have a higher prevalence of caries compared to girls, with the results being highly significant (*p* < 0.001). This finding was consistent with some studies [10,26], which could suggest that the mothers, especially in low socioeconomic status countries such as Egypt, tend to prefer and indulge boys, therefore they parent them differently. This is interpreted by giving them more sweets and hence they have a higher tendency for caries development [27].

Interestingly, the number of siblings in the family was a strong significant risk factor in the present study. The more siblings the child has, the more his or her risk to develop S-ECC. This goes in agreement with several studies, especially in families having more than two children [28,29]. One explanation for this is that in large families, parents’ attention towards their children’s oral health is shared or divided between the larger number of siblings, thereby less care is provided for each child and chances to have oral problems increase. Also, horizontal transmission may play a role in this domain.

The maternal and paternal educational level was inversely proportional to the presence of S-ECC in children. The results were similar to another study that attributed this finding to the lack of information and education about oral health care for children of minimally educated or uneducated parents [30,31]. However, the effect of paternal education seemed to be stronger as it probably reflects the socioeconomic level of the family [28].

In the present study, mothers’ current caries experience appeared to be an important factor in the development of S-ECC in children and the results were highly significant (*p* < 0.001). This was consistent with a study that reported evidence of maternal transmission of S. mutans in 41% of mother/child pairs [32]. A mother’s continuous contact with their children was interpreted by higher influence on child’s oral hygiene measures as compared to a father’s whose employment status, and education did not affect their children.

Children who had a late first dental visit or no dental visits at all tend to be more liable to develop dental caries. A possible interpretation for this could be that early dental visits provide an excellent opportunity for educating parents on proper guidelines for promoting their child’s oral health, especially among low socioeconomic groups where oral health is not considered a priority [33].

Regarding early feeding habits, the current study revealed no significant difference in caries experience between children who were breast or bottle fed. Our results were consistent with a systematic review revealing that the interaction of risk factors and intraoral bacterial load might carry a greater responsibility than the mode of feeding [28,34]. In addition, although children suffering from S-ECC were weaned at an older age than those in caries-free group, this difference was insignificant. Despite that, a significant association was detected in our research between overnight and on demand feeding and development of S-ECC. Night feeding decreases the clearance of liquid carbohydrates from the oral cavity due to decreased salivary flow at night. On the other hand, on demand feeding leads to increased amounts of fermentable carbohydrates in the mouth and early colonization by oral S. mutans [35]. Interestingly, this goes in line with a study attributing S-ECC mainly to the frequency of feeding rather than the age of weaning [36].

Considering other habits that could have a direct influence on bacterial transmission and caries incidence, sharing utensils was significantly more common in the S-ECC group as compared to the S-ECC free group. This explains why international guidelines for prevention of ECC [37] encourage parents to stop bacterial transmission to their children through sharing food, drinks, utensils, toothbrushes, and other items [32].

The sugary food intake frequency was higher among children with S-ECC than those who were caries-free. This was consistent with most of the literature which showed that caries incidence increases when the number of sugar containing snacks increased to more than four times a day [8]. Concerning the question regarding sugary snacks, it was explained to the caregivers to report any intake of sticky sweets as such as candies, marshmallows, and lollipops, etc. Although the nature of the food consumed played a role in the development of S-ECC, the scope of the study was not to categorize the sugar containing sweets, but rather to consider them as a single entity and prioritize how the frequency of their consumption ranked their weight as a risk factor for S-ECC. Children who consumed more sugary food, sweets, and sweetened beverages showed a significantly higher risk of developing S-ECC (*p* = 0.001) and this can be interpreted by the high sucrose content of confectionaries and sweetened beverages, since sucrose is considered the main carbohydrate responsible for the development of dental caries [33].

Among the beverages consumed that had a controversial relationship with the development of S-ECC in the literature were fruit juices [8]. In our study, there was a negative correlation between the consumption of 100% fruit juices and the presence of ECC (*p* < 0.001). On the other hand, artificially sweetened beverages most commonly contain sucrose or high fructose corn syrup, which are more effectively metabolized by Sm, while 100% juices contain fructose and glucose without sucrose [33].

From the results of this study it was shown that the ECC-free group adhered more to proper oral hygiene practices, especially the frequency of brushing (*p* < 0.001), regular brushing at night time (*p* = 0.005), as well as performance of the oral hygiene measures by the caregiver instead of the child himself (*p* = 0.001). Those results simulated a study, showing that parental training to brush their child’s teeth and daily frequency of tooth brushing are major determinants in S-ECC [38]. Another study stated that S-ECC is more common among children who commenced brushing at an age older than 24 months [24] and hence oral hygiene measures should be implemented as early as the eruption of the first tooth.

## 5. Conclusions

This study prioritized the risk factors contributing to severe early childhood caries and placed them in the order of their significance as follows: snacking of sugary food several times a day, increased number of siblings to three or more, night feeding, child self-employed brushing, mother’s caries experience, two siblings, on demand feeding, once/day sugary food, sharing utensils, one sibling, male gender, father’s education, late first child dental visit, brushing time, mother’s education, no dental visit, decreased brushing frequency, and no night brushing.

## Figures and Tables

**Figure 1 dentistry-05-00004-f001:**
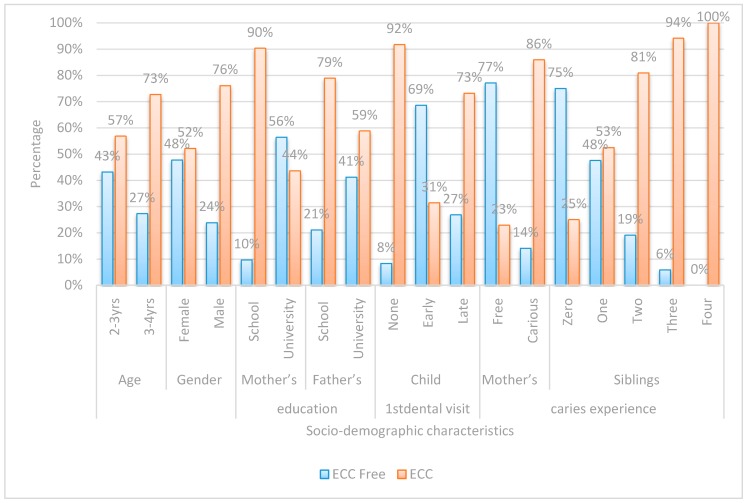
Bar chart comparing between early childhood caries (ECC)-free and children with ECC with regards to socio-demographic characteristics.

**Figure 2 dentistry-05-00004-f002:**
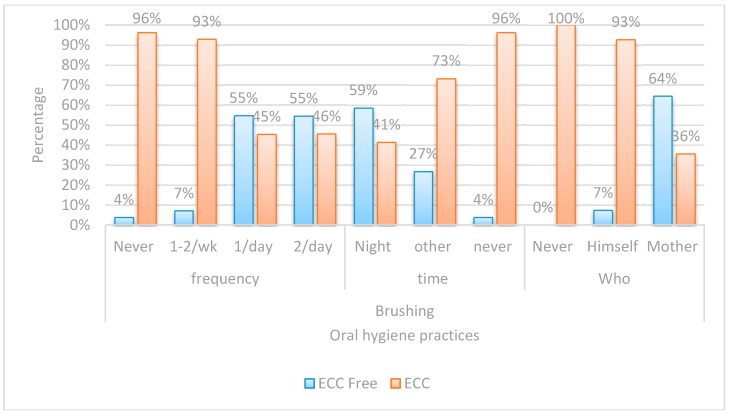
Bar chart comparing between children with ECC and ECC-free in regards to oral hygiene practice.

**Figure 3 dentistry-05-00004-f003:**
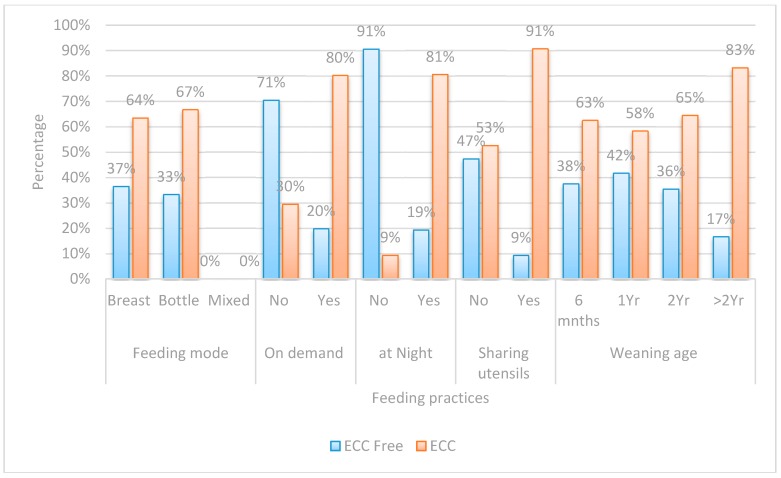
Bar chart comparing between children with ECC and ECC-free regards feeding practices.

**Figure 4 dentistry-05-00004-f004:**
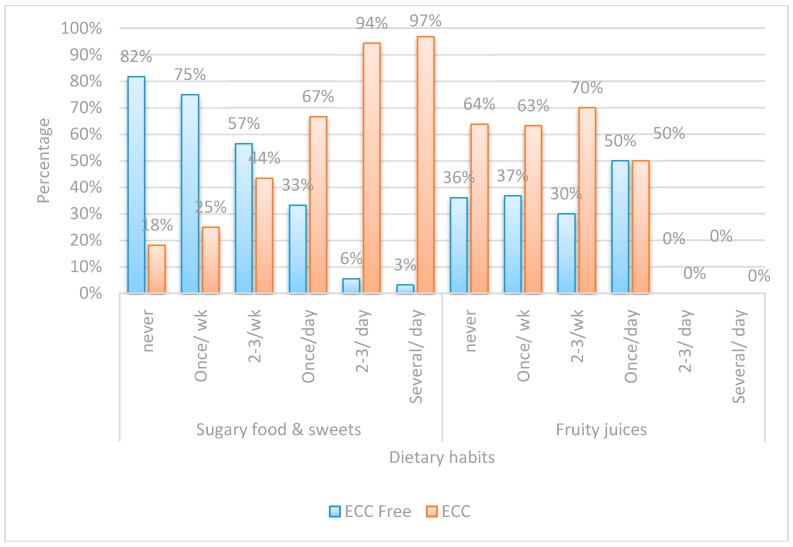
Bar chart comparing between children with ECC and ECC-free regards dietary practices.

**Table 1 dentistry-05-00004-t001:** Correlation between risk factors and log. Streptococcal count.

Risk Factors	Log. Streptococcal Count	
	Rho	*p*-Value
Age	−0.034	0.693
Male gender	0.287	0.001 **
Mother’s education	−0.483	<0.001 **
Father’s education	−0.198	0.019 *
Child’s first dental visit	−0.364	<0.001 **
Mother’s current caries experience	0.530	<0.001 **
Siblings	0.400	<0.001 **
Brushing frequency	−0.605	<0.001 **
Brushing time	−0.602	<0.001 **
Who brushes	−0.774	<0.001 **
Feeding on demand	0.452	<0.001 **
Night feeding	0.518	<0.001 **
Sharing utensils	0.420	<0.001 **
Sugary food intake frequency	0.584	<0.001 **

Spearman Correlations, ** HS = highly significant, <0.01 * S = significant. <0.05.

**Table 2 dentistry-05-00004-t002:** Logistic regression analysis of the most predictable risk factors associated with prevalence of ECC.

Risk Factors		*p*-Value	Exp. (β)	95.0% C.I. for Exp. (β)
Night feeding	No	Ref.		
	Yes	0.012 *	44.48	2.32–853.99
Mothers’ current caries experience	No	Ref.		
	Yes	<0.001 **	29.30	4.95–173.47
Gender	No	Ref.		
	Yes	0.017 *	11.54	1.56–85.54
Child’s first dental visit	None	0.022 *	Ref.	
	Late	0.961	1.069	0.073–15.74
	Early	0.031 *	0.068	0.006–0.78
Brushing time	Never	0.294	Ref.	
	Others	0.183	0.070	0.001–3.50
	Night	0.117	0.048	0.001–2.15
Brushing frequency	Never	0.424	Ref.	
	1/day	0.194	11.02	0.30–409.85
	2/day	0.552	1.72	0.29–10.18
Mother’s education	School	Ref.		
	University	0.175	0.094	0.003–2.87
Father’s education	School	Ref.		
	University	0.274	6.27	0.23–168.34
On demand feeding	No	Ref.		
	Yes	0.177	4.08	0.53–31.44
Sharing utensils	No	Ref.		
	Yes	0.648	0.607	0.71–5.16

** HS = highly significant, <0.01 * S = significant. <0.05; Ref = Reference.

**Table 3 dentistry-05-00004-t003:** Odd’s ratio of possible risk factors associated with prevalence of ECC (Master Table).

Risk Factor		ECC Free	ECC	OR	95% CI	*p*-Value
		(*n* = 50)	(*n* = 90)			
Frequency of Sugary Food Consumption	Never	18(81.8%)	4(18.2%)	Ref.		
	Several/Day	1(3.2%)	30(96.8%)	135.0	13.98–1303.95	<0.001 **
	Two to Three Times/Day	2(5.6%)	34(94.4%)	76.50	12.76–458.63	<0.001 **
Three Siblings	Zero	21(75.0%)	7(25.0%)	Ref.		
	Three	1(5.9%)	16(94.1%)	48.0	5.35–430.57	0.001 **
Night Feeding	No	29(90.6%)	3(9.4%)	Ref.		
	Yes	21(19.4%)	87(80.6%)	40.05	11.13–144.13	<0.001 **
Who Brushes	Mother	47(64.4%)	26(35.6%)	Ref.		
	Himself	3(7.3%)	38(92.7%)	22.90	6.43–81.48	<0.001 **
Mother’s Caries Experience	Free	37(77.1%)	11(22.9%)	Ref.		
	Carious	13(14.1%)	79(85.9%)	20.44	8.37–49.92	<0.001 **
Two Siblings	Zero	21(75.0%)	7(25.0%)	Ref.		
	Two	9(19.1%)	38(80.9%)	12.67	4.12–38.91	<0.001 **
On Demand Feeding	No	31(70.5%)	13(29.5%)	Ref.		
	Yes	19(19.8%)	77(80.2%)	9.66	4.26–21.93	<0.001 **
Sugary Food Once/Day	Never	18(81.8%)	4(18.2%)	Ref.		
	Once/Day	4(33.3%)	8(66.7%)	9.0	1.79–45.34	0.008 **
Sharing Utensils	No	46(47.4%)	51(52.6%)	Ref.		
	Yes	4(9.3%)	39(90.7%)	8.79	2.92–26.51	<0.001 **
One Sibling	Zero	21(75.0%)	7(25.0%)	Ref.		
	One	19(47.5%)	21(52.5%)	3.32	1.15–9.54	0.026 *
Gender	Female	33(47.8%)	36(52.2%)	Ref.		
	Male	17(23.9%)	54(76.1%)	2.91	1.42–5.99	0.004 **
Father’s Education	School	8(21.1%)	30(78.9%)	Ref.		
	University	42(41.2%)	60(58.8%)	0.381	0.16–0.91	0.030 *
Late First Child Dental Visit	None	4(8.3%)	44(91.7%)	Ref.		
	Late	11(26.8%)	30(73.2%)	0.248	0.072–0.852	<0.027 *
Brushing Time	Never	34(58.6%)	24(41.4%)	Ref.		
	Others	1(3.8%)	25(96.2%)	0.109	0.014–0.879	0.037 *
Mother’s Education	School	6(9.7%)	56(90.3%)	Ref.		
	University	44(56.4%)	34(43.6%)	0.083	0.03–0.21	<0.001 **
Early First Child Dental Visit	None	4(8.3%)	44(91.7%)	Ref.		
	Early	35(68.6%)	16(31.4%)	0.042	0.013–0.136	<0.001 **
Brushing Frequency	Never	1(3.8%)	25(96.2%)	Ref.		
	Once/Day	29(54.7%)	24(45.3%)	0.033	0.004–0.263	0.001 **
	Twice/Day	18(54.5%)	15(45.5%)	0.033	0.004–0.276	0.002 **
Brushing Time	Never	34(58.6%)	24(41.4%)	Ref.		
	Night	15(26.8%)	41(73.2%)	0.028	0.004–0.223	0.001 **

** HS = highly significant <0.01 * S = significant <0.05, Ref = odds ratio reference.

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
