# Peer review of "Prioritizing the Risk Factors of Severe Early Childhood Caries"

_dentistry, 2017, doi:10.3390/dj5010004_

Round 1

Reviewer 1 Report

The authors present an interesting research concept that would have interest to readers and oral health  scientists around the world. Having said this however, the presentation of the article needs significant assistance as follows:

1. While the English writing is fairly good there are significant problems in readability scattered throughout the article. The authors would benefit significantly from having a native English reader, perhaps most preferably someone with an oral health background, assist them in rewriting the article.

2. Layout of the article. Unless "Dentistry Journal" requires that the Results section come right out after the Introduction, the article should be re-organized such as that the Methods and Materials section is listed right after the Introduction followed by Results and then the Discussion.

3. At the end of the Introduction the last paragraph appears to attempt to describe the study's aim, purpose, and hypothesis; however, I do not believe that it does this very well and so would like to see that paragraph rewritten. In addition, why I like the use of the word reshuffling in the title, with in this paragraph the authors say they designed the study to reshuffle the order of significance of risk factors. I believe what they are actually attempting to do is just come up with an order of significance. I do not believe they have referenced any other research that attempted to list by order of significance the risk factors, so this is technically not a "reshuffling".

4. In the Results section, I struggle in understanding the definition of some of the items in their tables. For instance, under father and mother's education they say school and higher but give no definition for what exactly those two words mean in terms of the level of education or years of school or education, or etc. In addition, since this data was gathered through the use of a survey with the parents, it is important to recognize how poor reporting reliability can be from parents.

5. I could not find a reference for how the authors chose which risk factors to evaluate, that is parental education, who does the brushing, number of siblings, dietary definitions, etc. While I do not doubt that these are significant risk factors, it would have been good to clearly define why they chose the specific risk factors.

Unless I missed it, there is no section number 3. after the results section. The author skipped to section number 4.  materials and methods.

6. The Method section under plaque samples note that they did not collect plaque samples for all 140 children but gave no explanation of why. It seems as if they would have done this for all of the children? Also, the methodology they employed for the plaque samples is not referenced so it would be helpful to know why they used the method for sample collection and bacterial cultures.

7. Under item 4.8 feeding practices, it would have been good to define what they meant by sugary food, sweets and drinks.

Again, I believe there is an interesting data contained in this research project, but the article needs some work to get it into a publishable format. I hope the authors can take the time and find the expertise to help them do this. Thanks for the opportunity to review this interesting project.

Author Response

Dear Reviewer,

I would like to thank you for your meticulous reviewing.I have uploaded a new version of the manuscript highlighting the modified sections and will reply here separately to queries.

I took your advice and got  the manuscript revised by a native English reader and also made adjustments with the Grammarly program. Grammar & punctuation  changes are highlighted in yellow, while total rephrasing is highlighted in green.

The layout of the article has been modified and the numbering of the sections revised and modified.

The paragraph at the end of the introduction has been rewritten and the word reshuffling has been replaced by prioritizing as you pointed out the main scope was to list in the order of significance.

Definitions of levels of education have been clearly explained in the materials and methods section. We commented about self-reporting in the discussion and also that it was more accurate to use face to face interviewing as many of the caregivers did not have a high education level.

The review article from which we choose which factors to test has been referenced and mentioned in the discussion.We commented that although risk factors were almost 80 we choose those that were repeated and significant.

The method of bacterial sample collection has been rewritten and referenced. It should be noted that our study was self-funded, so it was not financially feasible for us to take samples from all participants, therefore a representative sample was taken and this was clarified in the materials and methods section.

As regards to the definition of sugary food, we clarified in the discussion section that it meant all sticky food like candies, lollipops etc.. and that's what was explained to the 

Finally, I would like to thank you again for your thorough review and apologize for the delay in reply as the modifications were major and hope you would re-evaluate our manuscript and find it eligible for publication.

Regards,

Noha Kabil. 

Acaregivers. It was also clarified in the discussion that we did not aim to categorize sugary food  but ,we regarded it as a whole entity concentrating upon the frequency of its consumption as our variable.

Reviewer 2 Report

Interestingly paper that re-prioritizes the risk factors that contribute to ECC. The tables can be presented in a better manner that makes it easier to read.  Also, minor corrections to spacing, punctuation, and capitalization are required.  For example, on line 43, there are periods in place of where commas should be.  In some lines, like 27 and 32, there are either too many spaces between words or no spaces at all.  Also, some sentences are lacking periods.  This occurs throughout the entire manuscript.  Overall, the findings presented in the paper will be useful for clinicians.

Author Response

Thank you for  your  review and the fruitful comments. I am glad that you found the English language and style to be fine and only needing minor corrections. I will correct all your remarks and upload a new edited version, with special consideration to punctuation, capital letters, and spacing and will rearrange the results section.

I really liked the word you chose about re-prioritizing and was thinking of changing the title to -prioritizing instead of reshuffling.

Finally, your opinion that our  findings will be useful for clinicians was very encouraging.

Regards,

 Noha Kabil.

Round 2

Reviewer 1 Report

I think the English reads much better, but there are still multiple sentences that don't flow smoothly and need rewriting. Otherwise, I believe the authors have for the most part address my suggestions to improve the article.

Author Response

Dear Reviewer, 

 My co-author and I are glad that we were able to address your suggestions.

The abstract was modified as well.

We added more information in the background, as you marked it as could be improved and rewritten several sentences, which we highlighted in green all through the manuscript after it was revised further by a native English speaker. 

Of course, nothing could be done at this point to improve the research design, but we replaced the first 4 tables by 4 figures or bar charts to make the results more clearly presented.

the old tables were left after the references section.

We are glad that you found the methods adequately described and that the conclusions are supported by the results.

I have uploaded the new modified version.

Thank you for your valuable input that has greatly upgraded our work.

Best regards,

Noha
